# Stability and Controller Research of Double-Wing FMAV System Based on Controllable Tail

**DOI:** 10.3390/biomimetics9080449

**Published:** 2024-07-24

**Authors:** Yichen Zhang, Yiming Xiao, Qingcheng Guo, Feng Cui, Jiaxin Zhao, Guangping Wu, Chaofeng Wu, Wu Liu

**Affiliations:** 1National Key Laboratory of Advanced Micro and Nano Manufacture Technology, Shanghai Jiao Tong University, Shanghai 200240, China; zhangyic@sjtu.edu.cn (Y.Z.); xiaoyiming0379@sjtu.edu.cn (Y.X.); guoqingcheng@sjtu.edu.cn (Q.G.); sdcuifeng@sjtu.edu.cn (F.C.); jiaxin.zhao@sjtu.edu.cn (J.Z.); excalibur996@sjtu.edu.cn (G.W.); 897124663@sjtu.edu.cn (C.W.); 2Department of Micro/Nano Electronics, School of Electronic Information and Electrical Engineering, Shanghai Jiao Tong University, Shanghai 200240, China

**Keywords:** wing–tail interaction, passive stability, FMAV, cascade controller

## Abstract

This study aimed to enhance the stability and response speed of a passive stabilized double-wing flapping micro air vehicle (FMAV) by implementing a feedback-controlled biomimetic tail. A model for flapping wings accurately calculated the lift force with only a 2.4% error compared to the experimental data. Experimental tests established the relationship between control torque and tail area, swing angle, and wing–tail spacing. A stability model for the double-wing FMAV was developed, incorporating stabilizing sails. Linearization of the hovering state facilitated the design of a simulation controller to improve response speed. By adjusting the feedback loops of velocity, angle, and angular velocity, the tail controller reduced the angle simulation response time from 4 s to 0.1 s and the velocity response time from 5.64 s to 0.1 s. In take-off experiments, a passive stabilized prototype with an adjustable tail angle exhibited enhanced flight stability compared to fixed tails, reducing standard deviation by 72.96% at a 0° take-off angle and 56.85% at a 5° take-off angle. The control axis standard deviation decreased by 38.06% compared to the passive stability axis, confirming the effectiveness of the designed tail angle controller in reducing angular deflection and improving flight stability.

## 1. Introduction

FMAVs achieve flight and hovering in the air by imitating the aerodynamics of insects or birds. They are characterized by their small size, agility, and high level of concealment [1,2]. They have garnered widespread attention from researchers because of their superior capability to execute reconnaissance and surveillance missions in narrow spaces and other extreme conditions [3,4]. Currently, researchers achieve the aerial hovering and free flight of FMAVs by emulating the flight patterns of two-winged insects, through the use of a control scheme that adjusts the trailing edge or alters the flapping plane [5,6].

Researchers have discovered, through studying the flight processes of birds with tails, that biomimetic tail surfaces can enhance damping effects. The combination of stabilizing surfaces achieved passive stability in prototypes. In this configuration, the center of gravity of the FMAV lies below the aerodynamic center, enabling the generation of control moments to maintain balance when the FMAV encounters disturbances [7]. Some researchers have successfully developed passive stable FMAV prototypes based on stabilizing sails [8,9,10,11]. However, these prototypes lack control mechanisms to adjust the oscillations and disturbances generated during flight. Rectification of these issues can be achieved by designing an appropriate active controller to adjust tail angles, leveraging the benefits of easily controlling passively stable prototypes.

Apart from assisting in self-balance, the angular deflection of a tail around the fuselage can generate additional control moments. Thomas found that birds often open and tilt their tails to one side when hovering. This suggests that the airflow generated by flapping induces flow around the tilted tail, resulting in dynamic pressure that exerts control moments on the tail [7]. Altartouri et al. also acknowledged the impact of induced flow due to the flapping on the tail [10]. Armanini et al. experimentally determined the wing–tail interaction, showcasing the intricate aerodynamics of the tail due to the induced flow from flapping wings [12]. Qian et al. replicated this phenomenon and effectively produced pitching moments by controlling tail deflection, showing alterations in flight attitude during their experiments [13]. Liang et al. developed an attitude feedback controller by adjusting the tail swing angle, thereby enabling the prototype to achieve hovering functionality [14,15]. Safeer et al. studied the sliding mode control of nonlinear systems [16]. These findings underscore the dual role of tail deflection in not just maintaining balance but also generating supplementary control moments, shedding light on its crucial contribution to achieving stable flight attitudes.

In addition to prototype experiments, establishing a theoretical model for tailed FMAVs helps enhance the flight performance of the prototypes. Some studies on flapping wing dynamics employ computational fluid dynamics (CFD) to simulate intricate fluid–structure interactions, necessitating substantial computational resources [17]. Quasi-steady models rely on derived or empirically determined force coefficients to model aerodynamic forces within a cycle, employing the method of equivalent average and rigid-body assumptions to simplify the model, saving time, and are suitable for high-frequency flapping scenarios [18,19,20,21]. Prototype system modeling often neglects the mass and inertia of wings, converting a multi-body dynamic model into a single-body dynamic model; then applying cycle averaging to transform nonlinear time-periodic dynamic models into nonlinear time-invariant models. This simplification facilitates a more straightforward design of controllers [22,23,24,25]. The modeling process for tail-equipped FMAVs is more intricate due to the influence of induced flow from flapping wings [26]. Wang et al. experimentally investigated the impact of changing the flapping frequency on the wing–tail interaction, to determine the optimal tail mounting position. An aerodynamic tail model and a suitable controller were designed for both simulation and physical prototypes in the hovering state to ensure stability [14,15]. It is evident from these analyses that experimental investigations are essential for understanding the influence of flapping-induced flow on the tail. Therefore, examining the interaction between flapping-induced flow and the tail under varied tail installation positions, areas, and inclination angles is necessary to establish effective tail and aircraft models, facilitating the subsequent design of an efficient controller.

In this study, a variable-angle biomimetic tail was designed to enhance stability during flight through the implementation of controllers. Initially, the lift model for the prototype’s flapping wings, the overall system dynamics model, and the force model of the tail induced by flapping wings were established. The wing mechanism was designed to test lift, ensuring model accuracy. Subsequently, the tail design was inspired by the wing–tail ratio observed in the ruby-topaz hummingbird during tail expansion. Experimental testing was conducted to determine the control moment induced by the tail due to the induced flow, fitting the relationship between tail control moment and tail area, installation position, and swing angle. In addition, the inertia and dimensional parameters of the prototype were obtained, and these parameters were used to determine the system model parameters. Active controllers were designed for simulation to assess optimized results relative to the original model. Finally, the controllers were applied to the physical prototype for take-off experiments, with the results compared to prior research. The validation confirmed that adjusting the tail angle during flight using the controller improved the prototype’s attitude stability.

## 2. Flap Lift Model

### 2.1. Description Movement

In order to describe the flapping wing model, a coordinate system delineated in Figure 1a–c was established. The inertial coordinate system $O_{i}X_{i}Y_{i}Z_{i}$ is affixed to the Earth’s frame of reference, with its $X_{i}$ axis oriented westward, the $Y_{i}$ axis facing southward, and the $Z_{i}$ axis pointing upwards. The origin of the fuselage coordinate system $O_{b}X_{b}Y_{b}Z_{b}$ is positioned at the fuselage’s center of gravity. The $X_{b}$ axis is aligned from the fuselage towards the back, the $Z_{b}$ axis extends from the tail to the nose, and the $Y_{b}$ axis conforms to the right-hand rule, proceeding from the left wing to the right wing. During the flapping wing’s hover, the fuselage coordinate system retains its orientation. The flapping wing coordinate system $O_{sp}X_{sp}Y_{sp}Z_{sp}$ is established, with the origin attached at the intersection of the wing root and the flapping wing axis. The $X_{sp}$ axis points backward from the fuselage, the $Z_{sp}$ axis moves forward from the tail of the aircraft, and the $Y_{b}$ axis direction is determined by the right-hand rule. The $X_{sp}O_{sp}Y_{sp}$ plane defines the flapping wing plane. The wing coordinate system $O_{w}X_{w}Y_{w}Z_{w}$ remains fixed with the wing, located at the wing root which coincides with the origin of the flapping wing coordinate system. The $X_{w}$ axis is directed below the wing, the $Y_{w}$ axis proceeds towards the tips of the wings along the leading edge of the wing, and the $Z_{w}$ axis orientation is also determined by the right-hand rule. Furthermore, it is assumed that there is passive twisting of the wings during the flapping process. Here, the flapping angle is denoted as ϕ, and the torsion angle is defined as $\alpha^{*}$. The formulas for flapping wings are presented in Equations (1) and (2).
(1)$$\phi =\phi_{0}+A_{m}\cos\left(2\pi ft\right)$$
(2)$$\alpha^{*}=\alpha_{0}+\frac{\alpha_{m}}{\tanh(k_{\alpha })}\tanh\left[k_{\alpha }\sin\left(2\pi ft\right)\right]$$
where $\phi_{0}$ represents the initial flapping angle of the flapping wing, $A_{m}$ denotes the flapping amplitude, *f* signifies the flapping frequency, $\alpha_{0}$ indicates the initial torsional deflection angle, and $\alpha_{m}$ represents the torsion amplitude. The parameter $k_{\alpha }$ influences the shape of the inclination function. Specifically, when $k_{\alpha }=0$, the torsion angle function forms a sinusoidal function, whereas a positive infinity for $k_{\alpha }$ results in a square wave for the torsion angle function.

The calculation of the wing’s motion necessitates exploring interrelationships among coordinate systems. Let us define ϕ as the reverse rotation angle around the Z axis of the wing coordinate system. The flapping wing coordinate system can then be obtained by reversing the rotation by an angle of 90+α around the Y axis, yielding the flapping coordinate system.

Furthermore, the angles that delineate the rotation of the fuselage coordinate system about the X axis, Y axis, and Z axis of the inertial coordinate system correspond to roll, pitch, and yaw. The rotation matrices around these three axes are given in Equations (3)–(5).
(3)$$R_{x}\left(\varphi \right)=\left[\begin{array}{ccc} 1 & 0 & 0\\ 0 & \cos\varphi & -\sin\varphi \\ 0 & \sin\varphi & \cos\varphi \end{array}\right]$$
(4)$$R_{y}\left(\theta \right)=\left[\begin{array}{ccc} \cos\theta & 0 & \sin\theta \\ 0 & 1 & 0\\ -\sin\theta & 0 & \cos\theta \end{array}\right]$$
(5)$$R_{z}\left(\psi \right)=\left[\begin{array}{ccc} \cos\psi & -\sin\psi & 0\\ \sin\psi & \cos\psi & 0\\ 0 & 0 & 1 \end{array}\right]$$

Then, the rotation matrix from the inertial coordinate system to the fuselage coordinate system is Equation (Equation 6).
(6)$$R_{IB}=R_{x}\left(-\varphi \right)R_{y}\left(-\theta \right)R_{z}\left(-\psi \right)$$

The rotation matrix from the flapping wing coordinate system $O_{sp}X_{sp}Y_{sp}Z_{sp}$ to the fuselage coordinate system $O_{b}X_{b}Y_{b}Z_{b}$ is Equation (Equation 7).
(7)$$R_{spb}=R_{z}\left(-\phi \right)$$

The position of the origin of the flapping wing coordinate system in the fuselage coordinate system is ${}^{b}{\vec{r}}_{\mathrm{bs}}=\left[\begin{array}{ccc} x_{w} & \pm e & z_{w} \end{array}\right]$, where e represents half the width of the fuselage, and $Z_{w}$ denotes the distance between the leading edge of the wing and the center of mass of the fuselage in the Z axis direction.

A planar geometry model of the wings was established, as shown in Figure 1d, where *R* is the wing length, *S* is the area of a single wing, *c* is the wing chord length, and $\bar{c}$ is the average chord length; then, $\bar{c}=S/R$. The aspect ratio of the wings is defined as $AR=2R/\bar{c}$. The chord length at r from the root of the wing is c(r), the leading edge function of the wing is $z_{1}\left(r\right)$, and the trailing edge function is $z_{2}\left(r\right)$; then, $c\left(r\right)=z_{1}\left(r\right)-z_{2}\left(r\right)$. The radius of rotation of the dimensionless k-order area moment of the wing is ${\hat{r}}_{\hat{\kappa }}={\left[\int_{0}^{1}\hat{c}\left(\hat{r}\right){\hat{r}}^{k}d_{\hat{r}}\right]}^{1/k}$. Deng et al. [27] solved the span of the center of pressure as Equation (Equation 8).
(8)$${\hat{r}}_{2}^{2}=\frac{\int_{0}^{R}r^{2}c\left(r\right)dr}{R^{2}S_{\mathrm{Wing}}}$$

The position of the center of pressure AC in the flapping coordinate system is ${}^{s}{\vec{r}}_{sw}=\left[\left(\frac{1}{4}-{\hat{x}}_{0}\right)\bar{c}\;\pm {\hat{r}}_{2}R\;0\right]$. ${\hat{x}}_{0}$ is the distance from the top of the wing to the axis of rotation of the wing. Assuming that the flapping wing shaft passes through the center of pressure, ${\hat{x}}_{0}$ is assigned a value of 0.25. The rotation matrix from the wing coordinate system $O_{w}X_{w}Y_{w}Z_{w}$ to the flapping wing coordinate system $O_{sp}X_{sp}Y_{sp}Z_{sp}$ is provided in Equation (Equation 10).
(9)$$R_{{}_{wsp}}=R_{{}_{y}}\left(-\frac{\pi }{2}-\alpha^{*}\right)$$

Illustrated in Figure 2, the velocity at the center of pressure of the wings encompasses the flapping motion of the wings along with the superposition of the spatial motion of the fuselage. The reference point (base point), $O_{sp}$, is employed, and the basis point method is applied to compute $U_{cp}$ as Equation (Equation 11).
(10)$$U_{cp}=R_{{}_{wb}}^{{}^{\prime }}\left(v+\omega \times r\right)+R_{{}_{wsp}}^{{}^{\prime }}\left(v_{{}_{wsp}}+\omega_{{}_{sp}}\times r_{{}_{sp}}\right)+\omega_{{}_{w}}\times r_{{}_{w}}+v_{{}_{w}}$$
where *v* denotes the speed of the fuselage, ω signifies the angular velocity of the fuselage, and *r* represents the distance from the origin of the fuselage coordinate system to the center of pressure, which can be calculated using Equation (Equation 12). $\omega_{w}$ signifies the angular velocity of the wing around the origin of the wing coordinate system $O_{sp}$, while $r_{w}$ indicates the distance from the center of pressure to $O_{sp}$. $v_{w}$ represents the translational velocity of the wings. $R_{sp}$ indicates the distance between $O_{sp}$ and $O_{b}$, which can be calculated using Equation (Equation 13).
(11)$$r=r_{w}+R_{wb}r_{c}$$
(12)$$r_{sp}=R_{wsp}r_{c}$$

The angular velocity of the flapping wing, denoted as $\omega_{sp}$, can be determined from the rotation matrix $R_{spb}$. In accordance with the coordinate system outlined in this paper, the equation representing the rotation matrix from the flapping wing coordinate system to the fuselage coordinate system is designated as $R_{wb}=R_{wsp}R_{spb}$. The flapping motion and torsional motion of the wings are expressed in the fuselage coordinate system as $\omega_{w}={\left[0,0,-\dot{\phi }\right]}^{T},\omega_{sp}={\left[\begin{array}{c} 0,-{\dot{\alpha }}^{*},0 \end{array}\right]}^{T}$.

### 2.2. Description of Force

A quasi-steady-state model is employed to characterize the lift force produced by the wings during the flapping process. This model encompasses three force components: the forces $F_{Ntr}$ and $F_{Ttr}$, acting in the normal and tangential translational directions of the wing; the force $F_{N}$, resulting from the wing’s rotation; and the force $F_{Na}$, arising from the inertia of the fluid with additional mass. The force stemming from the perturbation of airflow during wing flapping can be mathematically represented as Equation (Equation 13).
(13)$$F_{LTr}^{{}^{\prime }}=\frac{1}{2}\rho C_{L}\left(\alpha \right)cU_{cp}^{2}$$
where ρ is the air density, $C_{L}$ is the lift coefficient, and the force of the entire wing is given by the integral of the force section over the wingspan, as shown in Equation (Equation 14).
(14)$$F_{LTr}=\int_{0}^{R}F_{LTr}^{{}^{\prime }}\left(r\right)dr=\frac{1}{2}\rho C_{L}\left(\alpha \right)SU_{cp}^{2}$$

For resistance, a similar formula can be written (Equation (Equation 15)).
(15)$$F_{DTr}=\frac{1}{2}\rho C_{D}\left(\alpha \right)SU_{cp}^{2}$$

Given the ease of decomposing the force into the tangential and normal directions along the wing, the lift force is distinctly categorized as the tangential force $F_{NTr}$ and the normal force $F_{TTr}$, as Equations (16) and (17).
(16)$$F_{NTr}=\frac{1}{2}\rho C_{N}\left(\alpha \right)SU_{cp}^{2}$$
(17)$$F_{TTr}=\frac{1}{\gamma }\rho C_{\tau }\left(\alpha \right)SU_{cp}^{2}$$

This includes the effect of the delayed stall of the leading edge vortex, which is summarized as the lift coefficient $C_{L}$ and the drag coefficient $C_{D}$ [28] in Equations (18) and (19).
(18)$$C_{L}\left(\alpha \right)=0.225+1.58\sin\left(2.13\alpha -7.2\right)$$
(19)$$C_{D}\left(\alpha \right)=1.92-1.55\cos\left(2.04\alpha -9.82\right)$$

Additionally, a force arises from the rotation of the wing. This force is derived from the quasi-steady-state equation utilized in flutter analysis. Simultaneously, the rotating wing generates an added circulation around itself to satisfy the Kutta conditions. This phenomenon can be expressed as Equation (Equation 20) [29].
(20)$$\Gamma_{r}=\pi \alpha c^{2}\left(\frac{3}{4}-\overset{\wedge }{x_{0}}\right)$$

The force of two-dimensional airfoil is defined as Equation (Equation 21).
(21)$$F_{Nr}^{\prime }\left(r\right)=\rho U_{\infty }\Gamma_{r}=\pi \rho c^{2}\left(\frac{3}{4}-\overset{\wedge }{x_{0}}\right)U_{\infty }\alpha$$

The circulation-induced force acts usually at the center of pressure on the wing. The overall wing force is obtained by integrating across the wingspan as Equation (Equation 22).
(22)$$F_{Nr}=\int_{0}^{R}F_{Nr}^{\prime }dr=\pi \rho \left(\frac{3}{4}-\overset{\wedge }{x_{0}}\right)\alpha \phi R^{2}{\bar{c}}^{2}\int_{0}^{1}\overset{\wedge }{c^{2}}\left(r\right)\overset{\wedge }{r}d\overset{\wedge }{r}$$

$F_{nr}$ can be further reformulated using the center of pressure velocity $U_{cp}$, as in Equation (Equation 23).
(23)$$F_{Nr}=\pi \rho \left(\frac{3}{4}-\overset{\wedge }{x_{0}}\right)\dot{\alpha }U_{cp}\frac{R{\bar{c}}^{2}}{{\hat{r}}_{2}}\int_{0}^{1}\overset{\wedge }{c^{2}}\left(r\right)\overset{\wedge }{r}d\overset{\wedge }{r}$$

The force attributed to the inertia arising from the additional mass of the fluid can be mathematically expressed through equations initially introduced for flutter analysis. This force comprises two components: one pertaining to the apparent mass acceleration, and the other related to the centrifugal force. Previous research indicates that the impact of the additional mass’s inertia is relatively minor compared to the other two components, thus it can be disregarded for the sake of simplicity [29]. Consequently, the total force experienced by the flapping wing results from the summation of translation and rotation forces, as shown in Equations (24) and (25).
(24)$$F_{N}=F_{NTr}+F_{Nr}=\frac{1}{2}\rho C_{N}\left(\alpha \right)SU_{CP}^{2}+\pi \rho \left(\frac{3}{4}-{\overset{\wedge }{x}}_{0}\right)\dot{\alpha }U_{cp}\frac{R{\bar{c}}^{2}}{{\hat{r}}_{2}}\int_{0}^{1}\overset{\wedge }{c^{2}}\left(r\right)\overset{\wedge }{r}d\overset{\wedge }{r}$$
(25)$$F_{T}=F_{Tr}=\frac{1}{2}\rho C_{T}\left(\alpha \right)SU_{cp}^{2}$$

The distance between the center of pressure of the wing and the origin of the flapping coordinate system $O_{sp}$ is $r_{c}={\left[\left(\frac{1}{4}-x_{0}\right)\bar{c},{\hat{r}}_{2}R,0\right]}^{T}$.

The angle of attack of a wing refers to the vector representing the relative slope of the wing chord in relation to the surrounding air. It is defined as the projection of $U_{cp}$ onto the plane of the wing coordinate system XOZ. The aerodynamic component $U_{y}$ along the wingspan direction is not considered in this analysis as $U_{cp}=\sqrt{U_{x}^{2}+U_{z}^{2}}$. The angle of attack can be calculated as $\alpha =atan2(-U_{z},-U_{x})$. The function atan2 represents a four-quadrant arctangent with a return value ranging between −π and π. The time derivative of the angle of attack can be obtained from the aforementioned equation as Equation (Equation 26).
(26)$$\dot{\alpha }=\frac{{\dot{U}}_{z}U_{x}-U_{z}{\dot{U}}_{x}}{U_{z}^{2}+U_{x}^{2}}$$

The forces and moments experienced by the fuselage during the flight cycle are calculated as Equations (27) and (28).
(27)$${\left[X,Y,Z\right]}^{T}=\sum_{i}{\left[X_{i},Y_{i},Z_{i}\right]}^{T}=\sum_{i}R_{i}{\left[F_{Ti},0,F_{Ni}\right]}^{T}$$
(28)$${\left[L,M,N\right]}^{T}=\sum_{i}r_{ci}\times {\left[X_{i},Y_{i},Z_{i}\right]}^{T}$$

In the context of the fuselage coordinate system, X,Y, and *Z* represent forces along the respective X,Y, and *Z* axes. Correspondingly, L,M, and *N* denote moments around the *X* axis, *Y* axis, and *Z* axis of the fuselage. The theoretical model is validated by estimating the lift generated by insects in nature. Due to limited test data from bird-mimicking hummingbirds, and considering that the flight principles of hovering insects are consistent, insect flight data including from fruit flies [30], hoverflies [31], and moths [32] are used here for comparison. Table 1 provides the wingspan R, mean chord length $\bar{c}$, flapping frequency *f*, flapping amplitude $A_{m}$, and torsion angle $\alpha_{0}$ of different insect species. Average lift is computed using the above method and compared with results from related CFD analysis methods in Table 1. Parameters not listed in the table are provided based on a torsion angle amplitude $\alpha_{m}$ of 40°. The results indicate that this model demonstrates good accuracy when the flapping frequency approaches that of hummingbird flight. However, errors increase when the flapping frequency is too high, possibly due to unmentioned parameters.

To validate the accuracy of the lift model, a flapping mechanism was designed and constructed, and a lift testing platform was set up to test the lift results of the physical prototype. The wings of the flapping mechanism are 11 cm in length, and the flapping frequency of the wings was captured using a high-speed camera (Phantom VEO 710L, Vision Research, Wayne, NJ, USA). The mechanism was powered by a DC power supply (UTP1306S, UNI-T, Dongguan, China). The flap lift was measured using a Nano17 6-axis force sensor (manufactured by Industrial Automation, Raleigh, NC, USA, with a resolution of 0.3 gf) and the measurements were processed and outputted by the F/T Controller 9620-05-ctl-14 based on the sensor’s readings [33]. The test platform is shown in Figure 4. The sensor was calibrated with known force values in the Z direction before measuring lift, as per the calibration data shown in Table 2. The mean measurement error was found to be 0.26%, meeting experimental conditions. Different power output voltages were set, and the flapping frequency and lift values of the prototype were recorded. Figure 3a shows the curves of flapping theta and torsion theta within one cycle; Figure 3b illustrates the variation in the angle of attack (AOA) and angular of AOA throughout a single period. The observations reveal that the angle of attack curve aligns with the natural hovering motion of hummingbirds observed by Rivers [34], as shown in Figure 5a. Figure 3c documents the simulated lift and drag forces within one cycle; the lift patterns during the downstroke and upstroke are identical, while the drag forces act in opposite directions, consistent with the results indicated by the equations. The simulation results indicate a peak in the lift force due to the presence of the derivative term of the angle of attack in Equation (Equation 23), which represents the angle of attack velocity illustrated in Figure 3b. These findings are akin to Matej’s simulations [29], demonstrated in Figure 5b. The lift and drag coefficient trend depicted in the figure illustrates the lift and drag. Noteworthy are two peaks in both the lift and drag half-cycles, mirroring the simulation outcomes detailed in this paper. (The lift resistance profile within one period does not precisely match this paper, stemming from differing parameter conditions.) Since Matej’s work does not present the angular velocity, the sharp peak amplitude is lower. It is postulated that the angular velocity variation in the literature is less pronounced compared to this study. Various power output voltages were set to record the flapping frequency and lift values of the prototype by inputting the wing parameters of the flapping mechanism and the flapping frequency values of the test points into the simulation model to calculate theoretical lift, as shown in Figure 3d. Upon comparison, the average error between the experimental and simulated lift was found to be 2.4%, implying that the lift model can provide a fairly accurate estimation of the lift generated by the physical prototype.

## 3. Tail Design and Torque Measurements

### 3.1. The Morphology of a Hummingbird’s Tail

Designing the biomimetic tail of the FMAV based on a hummingbird’s characteristics is crucial. Birds’ tails play a significant role in maintaining stability within a certain range of flight speeds, generating lift and drag to aid in rolling, pitching, and yawing turns, as well as slow-speed flight. Different bird species often opt for varied tail shapes; however, they share a common trait—the aerodynamic function of the tail during flight—since a bird’s tail directly impacts stable flight [35] and can be utilized to maneuver body orientation [36] and ensure stable posture [37]. Sridhar Rav documented the extent to which a ruby-throated hummingbird opens its tail during the hunting process. This hummingbird enhances its stability by opening its tail while feeding, enabling it to hover effectively in the air [38]. The bird’s posture, as captured by the camera, is depicted in Figure 6. The illustration reveals that during hovering, the tail width is approximately 57% of the wingspan, and the tail length is around 1.63 times the chord length of the wings. Comparing these data with relevant studies in Table 3, Z. E. Teoh provided tail and wingspan dimensions [9], H. Altartouri offered wingspan and tail area information [10], and Breugel explained tail width and wingspan relationships [8]. The remaining parameters were estimated based on the proportions in research images.

### 3.2. Tail Force Model and Measurements

The induced flow velocity at the wing root is represented by u(0). This can be calculated from lift, wing area, and air density, as demonstrated in Wang’s work and further elaborated upon by Duhamel. et al. [39] through the momentum Equation (Equation 29).
(29)$$u_{i}\left(0\right)=\sqrt{\frac{F_{L}}{2S\rho }}$$

Drawing from the induced flow distribution law outlined by McCormick et al. [40], it is evident that the flow velocity along the body at $l_{t}$ from the leading edge is represented as Equation (Equation 30).
(30)$$u_{i}\left(l_{t}\right)=u_{i}\left(0\right)\left(1+\frac{\frac{l_{t}}{R}}{\sqrt{1+{\left(\frac{l_{t}}{R}\right)}^{2}}}\right)$$

In this equation, *R* represents propeller pitch for ornithopters in this paper. By utilizing the Bernoulli equation to calculate dynamic pressure, the aerodynamic force acting on the tail wing can be derived from Equation (Equation 31).
(31)$$F_{tail}=\frac{1}{2}\rho u_{i}\left(l_{t}\right)S_{tail}C_{l\alpha }\alpha_{T}$$

Parameter α denotes the tail pitch angle, typically designed with a flat profile. Under the assumption of a thin flat airfoil, the slope of the lift curve for a finite-span 3D wing is detailed in Equation (Equation 32) [41].
(32)$$C_{l\alpha }=\frac{2\pi }{1+\left(\frac{2L_{tail}}{D_{tail}}\right)\left(1+\tau \right)}$$

The values $L_{tail}$ and $D_{tail}$ represent parameters associated with the dimensions and shape of the tail, as explained in [41]. The ultimate pitch control moment produced by the tail is determined by the following Equation (Equation 33).
(33)$$T_{tail}=\frac{1}{2}\rho u_{i}\left(l_{t}\right)S_{tail}C_{la}\alpha_{T}x_{2}=N_{\alpha }\left(S_{tail},l_{t}\right)\alpha_{T}$$

Here, $N_{\alpha }$ represents the derivative of the moment with respect to the pitch angle of the tail, and $x_{2}$ is the distance between the pressure center of the tail and the center of gravity. The tail control torque relates to lt and $S_{tail}$, as well as $x_{2}$, and it can be determined through an experimental scheme. A schematic depiction of the tail control mechanism is presented in Figure 7a. Here, $d_{t1}$ refers to the connecting rod, while $d_{t2}$ represents the structure of the tail. $\theta_{t1}$ denotes the limiting angle of the servo’s rocker arm, and $\theta_{t2}$ signifies the angle between the tail structure and the horizontal axis. The rotational angle of the tail must initially be ascertained. Additionally, $d_{ta}$ indicates the servo’s rocker arm, and $d_{tb}$ is the distance from the top of the wing to the servo rocker. The targeted parameters for the tail are detailed in Table 4, and the dimensions of $d_{t1}$ and $d_{t2}$ are tailored to fulfill the movement objectives.

From the positional arrangement of the points illustrated in Figure 7a, the coordinate parameters of each point can be derived as $A_{l}=\left(d_{ta}\cos\theta_{tl},d_{ta}\sin\theta_{tl}\right),B_{l}=\left(d_{t2}\cos\theta_{t2},-d_{tb}-d_{t2}\sin\theta_{t2}\right),A_{2}=\left(d_{ta}\cos\theta_{tl},-d_{ta}\sin\theta_{tl}\right),B_{l}=\left(0,-d_{tb}-d_{t2}\right)$ The distance relationship is obtained according to the coordinate relationship, as in Equations (34) and (35).
(34)$$A_{1}B_{1}=d_{t1}=\sqrt{{\left(d_{ta}\cos\theta_{t1}-d_{t2}\cos\theta_{t2}\right)}^{2}+{\left(d_{ta}\sin\theta_{t1}-\left(-d_{tb}-d_{t2}\sin\theta_{t2}\right)\right)}^{2}}$$
(35)$$A_{2}B_{2}=d_{t1}=\sqrt{{\left(d_{ta}\cos\theta_{t1}-d_{t2}\cos\theta_{t2}\right)}^{2}+{\left(-d_{ta}\sin\theta_{t1}-\left(-d_{tb}-d_{t2}\right)\right)}^{2}}$$

The amplitude of the steering gear swing angle is 45°, and as per the solution to the joint equation, $d_{t1}$ = 2.88 cm and $d_{t2}$ = 1.21 cm. In order to investigate the varying relationship of $T_{tail}$ with $S_{tail}$, $l_{t}$, and $\alpha_{t}$, an experimental platform for measuring servo torque was constructed. This platform comprises two main sections: the upper segment features a height-adjustable flapping wing mechanism used to induce flow, while the lower part includes a tail and a load cell for testing control torque. The experimental setup is depicted in Figure 7b.

The test bench comprises an upper platform housing a vertical chute, a flapping wing mechanism, a tail, an F/T measuring device, and a fixed base. The flapping wing mechanism is affixed on the upper platform and can move along the vertical chute. The tail is secured to the force-measuring device and fixed to the base. Throughout the measurement process, adjustments were made to the distance between the flapping wing mechanism and the tail. Tests were conducted at a 24 Hz flapping frequency, maintaining consistency in the induced flow velocity for each experiment. The dynamometer recorded the moment experienced by the tail.

To explore different areas, two tails with dimensions of 8 cm × 3 cm, 8 cm × 5 cm, and 8 cm × 8 cm were constructed for testing. The physical representation of the tail, shown in Figure 8a, consists of a carbon fiber prepreg skeleton and PET used as the tail film, arranged symmetrically on the left and right sides, as illustrated in Figure 7c. When rotated in the same direction, the pitch axis control torque is generated, while turning the tail in the opposite direction generates the yaw axis control torque. The tail is manipulated by a servo, and its angle is altered using a 4-link mechanism.

During the experiment, the tail was initially installed on the sensor, and the servo rotation controlled the tail within $45^{\circ }$ amplitude, while driving the tail to rotate within a $\pm 35^{\circ }$ range. For testing the control torque of the pitch axis, the servo rotated the two tails in the same direction, and for testing the yaw axis torque, the two tails were rotated in opposite directions.

The flapping wing mechanism commenced flapping at a frequency of 24 Hz, maintained at distances of 5 cm, 7 cm, and 9 cm from the top of the tail, respectively. The sensor measured the torque at different angles of the tail. The sampling frequency of the force measuring device was 3000 Hz. By adjusting the tail’s mounting to keep the distance from the center of the tail to the sensor constant ($l_{measure}$ = 5 cm), upon observation, it was found that the servo motor took about 5.9 s to complete one rotation, resulting in approximately one observation point being captured every 1.47 s.

By adjusting the tail installation to ensure a constant distance ($l_{measure}$) of 5 cm from the sensor, the measurement results shown by the blue curve in Figure 8b were obtained. Although the raw data exhibit some increasing pattern, they fluctuate significantly. The standard deviation of the regression value errors calculated using linear regression methods is 3.9256 N·mm, making the direct use of raw data impractical. These errors are believed to be caused by the vibrational noise from the periodic flapping motion. Therefore, noise reduction can be achieved by applying an average smoothing filter. Using a window of 1000 data points, the filtered data are represented by the red curve in Figure 8b. The standard deviation of regression value errors calculated using linear regression on the filtered data is 0.782 N·mm, effectively capturing the torque variation trends at different angles. Therefore, the filtered data are preferred for the fitting analysis, after being processed. During actual flight, it is assumed that $S_{tail}$ and $l_{t}$ remain constant, while the relationship between $T_{tail}$ and $\alpha_{T}$ is considered crucial for control research. This relationship is delineated by Equation (Equation 36), obtained through linear fitting, illustrated by the comparison between the fitted data and raw data in Figure 8c.
(36)$$T_{tail-measure}=0.08\left(0.005S_{tail}-0.053l_{t}\right)\alpha_{T}$$

The fitting results reveal that the measured moment is directly proportional to $S_{tail}$ and inversely proportional to $l_{t}$, aligning with the principle of air flow attenuation. The standard deviation of the fit measures 0.088, and the fitting error for each measurement point is depicted in Figure 5b. These results establish that the fitting equation effectively represents the actual outcomes. Given the difference in the moment application point between actual flight and measurement, an equivalent effect method is employed to compute the tail moment during actual flight. This equivalence can be calculated as follows (Equation (Equation 37)):(37)$$F_{tail_{e}q}=\frac{T_{tail_{m}easwe}}{l_{measure}}$$

The actual moment in flight is given by Equation (Equation 38).
(38)$$T_{tail}=-F_{tail_{-}eq}l_{tail_{-}c}$$

$l_{tail\_c}$ represents the distance from the tail’s center to the center of mass. As the measured moment opposes the control torque during flight, a negative sign should be included. Based on the author’s previous research, a tail width of 16 cm meets passive stability requirements [42]. To mimic natural hummingbird morphology, a tail length of 5 cm was chosen, as this dimension closely corresponds to the aspect ratio of the chord length found in hummingbirds.

## 4. FMAV Drag Model of Angle-Variable Tail

In accordance with the quasi-steady-state aerodynamics theory, the air resistance encountered on the airfoil surface can be represented by Equation (Equation 39).
(39)$$F_{D}=\frac{1}{2}\rho C_{D}\left(\alpha \right)S_{tail}v^{2}=kv^{2}$$
where ρ represents the air density, *S* denotes the wing area, $C_{D}$ signifies the drag coefficient associated with the average angle of attack, and *v* symbolizes the air velocity atop the wing surface, with all constant terms simplified to *k*. Considering the free air velocity *u* and the relative wing velocity *w* in relation to the fuselage, the air velocity during the downstroke is v=w+u. In contrast, during the upstroke, as the wing moves considerably faster than the free airflow velocity, the air resistance acts in the opposite direction, resulting in an air velocity of v=w−u. Given that the upstroke and downstroke have equal durations, the average stroke force can be expressed as Equation (Equation 40).
(40)$$\bar{f_{d}}=\frac{1}{2}k\left[-{\left(w+u\right)}^{2}+{\left(w-u\right)}^{2}\right]=-2kwu$$

The angular velocity w is linked to the flapping frequency; we can infer that air resistance maintains a linear relationship with flow velocity. Here, we set the damping factor to b=2kw. As the flapping wing robot approaches hovering, its dynamics can be approximately described as a rigid body. Furthermore, the longitudinal and transverse dynamics are weakly coupled and can be elucidated using the linearized Newton–Euler equations. In order to simulate the hovering state, we assume that θ is small and that the vertical velocity vz is also small. To induce lateral movement in response to aerodynamic drag, the sail is stabilized perpendicular to the direction of velocity. Within the model, dampers are set both above and below the aircraft’s center of mass. When the aircraft undergoes lateral movement or tilts, wind resistance applies force and corresponding torque on the center of mass, as depicted in Figure 9.

When the aircraft undergoes lateral movement, it is influenced by aerodynamic drag ($F_{1},F_{2},F_{b}$), gravity (mg), and the thrust ($F_{L}$) generated by the flapping wing. The drag is a result of the aircraft’s translational velocity *v* and the angular velocity ω related to rotation around the center of mass. The equation corresponding to the horizontal direction is given by Equation (Equation 41).
(41)$$\begin{array}{cc} m\dot{\nu } & =F_{1}+F_{2}+F_{b}+F_{Lx}\\ & =-b_{1}\left(\nu -\omega d_{1}\cos\theta \right)-b_{w}\left(\nu -\omega d_{w}\cos\theta \right)\\ & -b_{2}\left(\nu +\omega d_{2}\cos\theta \right)\cos\alpha_{T}+F_{L}\sin\theta \end{array}$$

$d_{1}$, $d_{2}$, and $d_{w}$ denote the distances between the point of action of aerodynamic drag and the center of gravity of the mobile air vehicle (MAV) with dampers, while $b_{1},b_{2}$, and $b_{w}$ represent the corresponding aerodynamic damping coefficients. Here, *m* refers to the mass of the aircraft, inclusive of the damped sail. Assuming that the thrust $F_{l}$ aligns with the center of gravity, the FMAV (with damper) solely experiences the torque arising from the aerodynamic drag as Equation (Equation 42).
(42)$$\begin{array}{cc} J\dot{\omega } & =T_{1}+T_{w}+T_{2}+T_{tail}\\ & =-F_{1}d_{1}-F_{w}d_{w}+F_{2}d_{2}+T_{tail}\\ & =b_{1}d_{1}\left(v-\omega d_{1}\cos\theta \right)+b_{w}d_{w}\left(v-\omega d_{w}\cos\theta \right)\\ & -b_{2}d_{2}\left(v-\omega d_{2}\cos\theta \right)\cos\alpha_{T}+N_{\alpha }\alpha_{T} \end{array}$$

In this context, *J* signifies the moment of inertia of the FMAV (with damper). Assuming that the center of lift and the center of gravity align vertically, the lift $F_{l}$ does not generate a moment. During hovering, when the pitch axis declination angle θ is exceptionally small, we can approximate cosθ to be approximately 1 and sinθ to be approximately 0. According to Liang Wang [14,15], the influence of the servo is also considered to be a linear system for analysis; therefore, we hypothesize it to be approximately 1, and further, we assume the lift $F_{l}\approx mg$. For the sake of facilitating attitude stability analysis, we consider the velocity *v*, pitch angle θ, and angular velocity ω as state variables. Thus, we construct the state vector $x={\left[v,\theta ,\omega \right]}^{T}$. This approach allows us to reframe the kinetic Equations (41) and (42) as represented by Equation (Equation 43):(43)$$\left\{\begin{array}{l} =\frac{1}{m}\left[\left(-b_{1}-b_{2}-b_{w}\right)v+\left(b_{1}d_{1}-b_{2}d_{2}+b_{w}d_{w}\right)\omega +mg\theta \right]\\ =\omega \\ =\frac{1}{J}\left[\left(b_{1}d_{1}+b_{w}d_{w}-b_{2}d_{2}\right)v+\left(-b_{1}d_{1}^{2}-b_{w}d_{w}^{2}-b_{2}d_{2}^{2}\right)\omega +T_{tail}\right] \end{array}\right.$$

Equation (Equation 43) can be rewritten in the form of a state-space equation as Equation (Equation 44).
(44)$$\left[\begin{array}{c} \dot{v}\\ \dot{\omega }\\ \dot{\theta } \end{array}\right]=\left[\begin{array}{ccc} X_{u}^{+} & X_{q}^{+} & g\\ M_{u}^{+} & M_{q}^{+} & 0\\ 0 & 1 & 0 \end{array}\right]\left[\begin{array}{c} v\\ \omega \\ \theta \end{array}\right]+\left[\begin{array}{c} 0\\ N_{\alpha }^{+}\\ 0 \end{array}\right]\alpha_{T}$$
while $X_{u}^{+}=\frac{1}{m}\left(-b_{1}-b_{2}-b_{w}\right),X_{q}^{+}=\frac{1}{m}\left(b_{1}d_{1}-b_{2}d_{2}+b_{w}d_{w}\right),M_{u}^{+}=\frac{1}{J}\left(b_{1}d_{1}+b_{w}d_{w}-b_{2}d_{2}\right),M_{q}^{+}=\frac{1}{J}\left(-b_{1}d_{1}^{2}-b_{w}d_{w}^{2}-b_{2}d_{2}^{2}\right),N_{a}^{+}=\frac{1}{J}N_{a}$. The pitch axis dynamics equation is modified into regions, as $\dot{x}={\left[\dot{v},\dot{\omega },\dot{\theta }\right]}^{T},x={\left[\begin{array}{c} v,\omega ,\theta \end{array}\right]}^{T},B={\left[0,N_{\alpha }^{+},0\right]}^{T},K=\alpha_{T},A=\left[\begin{array}{ccc} X_{u}^{+} & X_{q}^{+} & g^{+}\\ M_{u}^{+} & M_{q}^{+} & 0\\ 0 & 1 & 0 \end{array}\right]$. Through analyzing the eigenvalues of matrix A, passive stability of the model can be discerned. If all eigenvalues reside to the left of the dashed axis, the system is stable. When the maximum eigenvalue is situated on the imaginary axis, the system is critically stable. However, if there are eigenvalues present to the right of the dashed axis, the system is deemed unstable. The equation is modified to the standard type of transfer function matrix $X={\left(sI-A\right)}^{-1}Bk$, and meanwhile, the corresponding transfer function by ${\left(sI-A\right)}^{-1}=\frac{{\left(sI-A\right)}^{*}}{\left|sI-A\right|}$ is obtained as Equations (45) and (46).
(45)$$\frac{U(s)}{\alpha_{T}\left(s\right)}=\frac{N_{\alpha }^{+}\left(X_{q}^{+}s+g^{+}\right)}{s^{3}-\left(M_{q}^{+}+X_{u}^{+}\right)s^{2}+\left(X_{q}^{+}M_{u}^{+}-M_{q}^{+}X_{u}^{+}\right)s-M_{u}^{+}g}$$
(46)$$\frac{Q(s)}{\alpha_{T}\left(s\right)}=\frac{N_{x}^{+}\left(s^{2}-X_{u}^{+}s\right)}{s^{3}-\left(M_{{}_{q}}^{+}+X_{{}_{u}}^{+}\right)s^{2}+\left(X_{{}_{q}}^{+}M_{{}_{u}}^{+}-M_{{}_{q}}^{+}X_{{}_{u}}^{+}\right)s-M_{{}_{u}}^{+}g}$$

The model parameters are derived from the prototype parameters, and the controller is designed using the root trajectory method, ultimately enhancing the response performance of the prototype.

## 5. Simulation Model Controller Design

### 5.1. Analysis of Model Stability

The physical prototype is depicted in Figure 10, and the essential parameters are outlined in Table 5. This prototype comprises a stationary upper sail, a flapping wing mechanism, and an adjustable lower sail capable of altering its angle. The lower sail is controlled by a steering gear, which facilitates the angle adjustment through the connecting rod mechanism illustrated in Figure 7a. Fluorescent labeled balls are affixed to the prototype for motion capture purposes. Based on the prototype parameters, the model parameters can be computed, as presented in Table 6.

Substituting the physical prototype parameters into the model matrix yields eigenvalues of $\lambda_{1}=-14.76,\lambda_{2}=-0.56+4.51i$, and $\lambda_{3}=-0.56-4.51i$. This indicates that all the real parts of the eigenvalues are less than 0, signifying that the model is in a passively stable state.

### 5.2. Series Simulation Controller Design

The control scheme for the model employed the series feedback control method, a technique which has been repeatedly utilized for stable control of FMAVs with consistently good results [29,43]. A block diagram representing the feedback control model is illustrated in Figure 11. The control module comprises a total of three layers of feedback loops. The innermost layer is devoted to the angular velocity loop. If the sole objective is to regulate angular velocity, adjustment of the control parameters within the angular velocity loop suffices. The middle layer constitutes an angle loop. When controlling the angular velocity along with angle regulation, simultaneous determination of the control parameters for both the angular velocity and angle loops is necessitated. The outermost layer covers the linear velocity loop. To manage and regulate linear velocity, the parameter determination process for all loops must be completed. In this paper, parameter adjustments are carried out in the order of loop-*q*, loop-θ, loop-*v*.

The adjustment of the angular velocity loop (loop-q) applies the root trajectory method. The root trajectory image of the open-loop transfer function $\frac{Q(s)}{\theta_{tail}\left(s\right)}$ is depicted in Figure 12a. In this figure, the trajectory of the root signifies the model’s root change as the proportional gain varies, with a minimum gain value selected at 6.9.

The unit step response curve of the closed-loop transfer function is illustrated in Figure 12b, indicating a significant steady-state error in the model when applying only proportional adjustments, necessitating the introduction of an integral term. By tuning the integral term coefficient, the final steady-state error is reduced to less than 0.02, with the integral term coefficient set at 700. In a similar fashion, the angle loop (loop-θ) is adjusted. Figure 12c illustrates the open-loop root trajectory image of the angle loop, confirming a proportional gain coefficient of 4390. Subsequently, the closed-loop unit step response curve displayed in Figure 12d shows a steady-state error of approximately 0, signifying that the proportional adjustment rendered the system stable without requiring additional control.

Following this, the linear velocity loop undergoes adjustment, where the open-loop root trajectory image of the linear velocity loop is shown in Figure 12e, ultimately resulting in the determination of a proportional gain coefficient of 14.5. The unit closed-loop step response curve is depicted in Figure 12f, revealing an approximate 2% steady-state error, indicating the successful achievement of the controller effect.

To demonstrate the controller’s impact, the model’s original response is compared with that of the model incorporating the controller, focusing specifically on the angular response shown in Figure 12g. The uncontrolled model exhibits a maximum angular overshoot of 47.11%, with a result error adjustment time of less than 5% of the steady-state value, equating to 4 s. After integrating the control system, there is no longer any overshoot, and the adjustment time becomes 0.1 s. Consequently, the simulation results indicate a significant enhancement in system response due to the controller. Subsequently, the controller’s performance is verified using a physical prototype.

## 6. Physical Prototype Testing and Validation

The prototype depicted in Figure 10 was levitated to authenticate the efficacy of the controller. The Vicon system (Vicon Motion Systems, Oxford, UK) was utilized to record the prototype’s orientation during flight. With a capture precision of 0.2 mm and a frequency of 100 Hz [44], the Vicon system is widely used in aircraft attitude capture experiments, as explained by Kim [44]. Four spherical reflective markers were carefully positioned on the prototype to meet asymmetrical motion capture demands. The prototype utilizes the main control board [45], embedded with an MPU9250 gyroscope (InvenSense, San Jose, CA, USA) to gauge its state. A closed-loop cascade controller PID algorithm was constructed to stabilize the prototype’s flight orientation by managing the tail swing angle. Owing to the prototype’s inherent passive stability, the PID parameters can be adjusted during flight. A take-off platform, as illustrated in Figure 13, was constructed. To assess the prototype’s resistance to interference, the platform was set at inclination angles of 0° and 5°, demonstrating the capability to take-off and maneuver by documenting the prototype’s lift-off process.

Figure 14 depicts a video clip of the controlled take-off experiment at a 0° tilt, showcasing the testing conditions and the flight process. In the video, a picture of the flapping prototype is captured every 3 s, focusing on enlarging a specific area of the FMAV. Flight data results are presented in Figure 15. Figure 15a illustrates the flight path of a prototype using closed-loop control with zero tilt on the take-off platform. Figure 15b displays the pitch and roll axis angles of the prototype during flight. The average pitch axis inclination of the prototype measured 0.0661 rad, with a standard deviation of 0.0663 rad, while the roll axis inclination recorded −0.1185 rad, with a standard deviation of 0.0275 rad. Figure 15c showcases the flight path of the prototype under open-loop no-feedback control conditions without any platform tilt, while Figure 15d portrays the pitch and roll axis angles during flight. The average pitch inclination angle registered at 0.1842 rad with a standard deviation; the roll axial declination was −0.0396 rad, with a standard deviation measuring 0.1525 rad. Figure 15e outlines the flight trajectory of the prototype employing closed-loop feedback control, with a 5° tilt on the take-off platform. Correspondingly, Figure 15f depicts the pitch and roll axes during flight. The average pitch inclination and standard deviation measured 0.0434 rad and 0.0489 rad, respectively, while the roll axial declination was −0.0583 rad, accompanied by a standard deviation of 0.0395 rad. Figure 15g depicts the flight path of the prototype under open-loop no-feedback control conditions, with a 5° tilt on the take-off bench, whereas Figure 15h illustrates the pitch and roll axis angles during flight. Under both take-off conditions, the discernible angular stabilization through closed-loop control was apparent. Specifically, under the 0° take-off condition, the standard deviation for pitch direction reduced by 64.00%, and for the roll direction, it decreased by 81.97%. Meanwhile, under the 5° take-off circumstance, the standard deviation of the pitch direction lessened by 56.53%, and that of the roll direction lowered by 57.16%. These experimental findings mirror the simulation outcomes, verifying the prototype’s improved flight attitude stability through closed-loop feedback control. The comparative flight videos can be viewed in the Appendix A.

In previous work, the authors designed a passive stable double-wing prototype without control mechanisms [42]. A comparison is made between the test results of this paper and the earlier study. The original data testing results are shown in Table 7. At a 0° take-off angle, the standard deviation in the pitch direction in this paper is reduced by 40.16% compared to the previous work (average standard deviation of 0.1127 rad), and the standard deviation in the roll direction decreases by 82.88% compared to the previous work (average standard deviation of 0.1606 rad). When taking off at a 5° tilt, the standard deviation in the pitch direction in this paper is reduced by 61.24% compared to the previous work (average standard deviation of 0.126 rad), and the standard deviation in the roll direction decreases by 75.16% compared to the previous work (average standard deviation of 0.1590 rad). The deviation level in this paper without feedback processes shows no significant difference compared to the previous work, indicating that a feedback-controlled tail can reduce attitude jitter during hovering.

This paper achieved stability control on the pitch and yaw axes of the biomimetic tail, enhancing flight stability under passive conditions of the prototype. However, compared to three-degree-of-freedom control in tailless flight, the prototype still cannot achieve stable flight under passive conditions. Therefore, our next step is to implement three-degree-of-freedom flight control for tail adjustment, thereby achieving stable flight for passive unstable prototypes and expanding the range of application scenarios for tail control solutions.

## 7. Results

This study primarily investigates enhancing the flight stability of passive stable bi-winged FMAVs by controlling the swing angle of a biomimetic tail. We applied the sliding element method to construct a double-wing flapping wing lift model using quasi-steady-state theory. Our simulation results demonstrated the average error of the lift model is 2.4% at various frequencies, effectively achieving the desired outcome. Based on the proportional relationship between the wing span and tail observed during feeding by ruby-topaz hummingbirds, the size of the tail was determined. Experimental investigations were carried out to establish the link between the flapping wing-induced flow moment on biomimetic tail and its area, wing–tail spacing, and tail swing angle. We proceeded by designing and fabricating a physical prototype of a double-winged flapping wing comprising upper and lower damping sails. Subsequently, we established a comprehensive system model for the entire aircraft using prototype size parameters and inertia parameters, thereby calculating the model’s characteristic values to confirm its passive stability. During the simulation, we developed a series PID controller and determined the specific controller parameters using the trajectory method for the angular velocity inner loop, angle intermediate loop, and velocity outer loop in sequence. After the implementation of the controller, the model’s angle decreased from 0.1 rad to zero, the settling time reduced from 4 s to 0.1 s, and overshoot was eliminated. Similarly, the linear speed decreased from 0.1 m/s, with the settling time plummeting from 5.64 s to 0.1 s, effectively removing overshoot. This same approach was implemented in the physical prototype to achieve closed-loop control. To validate the efficacy of tail inclination adjustment through the controller for enhancing the passively stable attitude stability of the FMAV, take-off experiments were conducted. Under the initial 0° condition, the feedback-adjusted prototype displayed a reduction of 64.00% in standard deviation along the pitch axis and 81.97% along the roll axis direction. Under a 5° take-off condition, the pitch direction standard deviation decreased by 56.53%, and the roll direction decreased by 57.16%. In comparison with the author’s previous work, under the initial condition of a 0° pitch angle, feedback control reduced the standard deviation of the pitch axis by 40.16% and that of the roll axis by 82.88%; under the take-off condition of a 5° pitch angle, the standard deviation decreased by 61.24% in the pitch direction and by 75.16% in the roll axis direction. These results serve to verify that controller-based tail inclination adjustments are beneficial for enhancing the passive stability of an FMAV’s attitude.

## Figures and Tables

**Figure 1 biomimetics-09-00449-f001:**
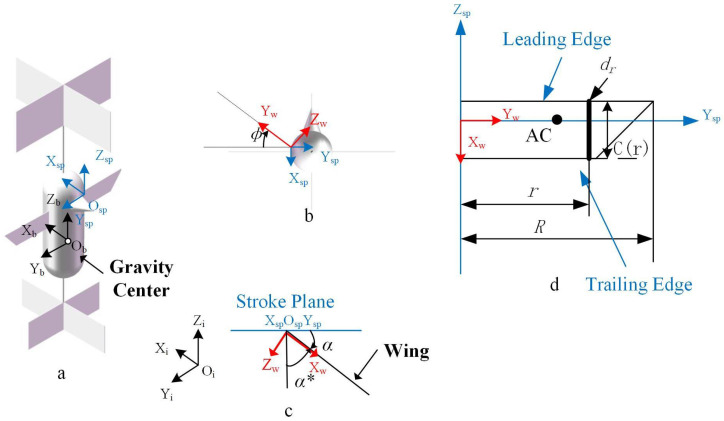
Coordinate system of the insect-like aircraft: (**a**) isometric diagram of the whole aircraft model; (**b**) top view; (**c**) schematic diagram of angle of attack and inclination angle; (**d**) schematic diagram of wing parameters.

**Figure 2 biomimetics-09-00449-f002:**
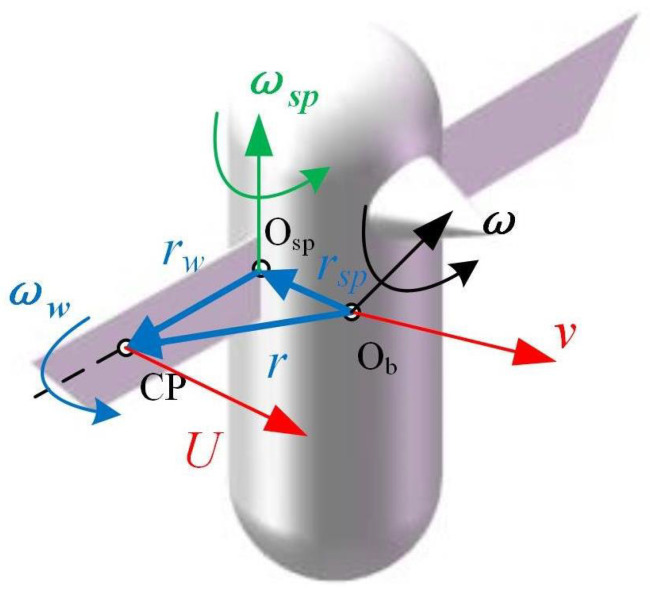
Schematic diagram of linear velocity and angular velocity.

**Figure 3 biomimetics-09-00449-f003:**
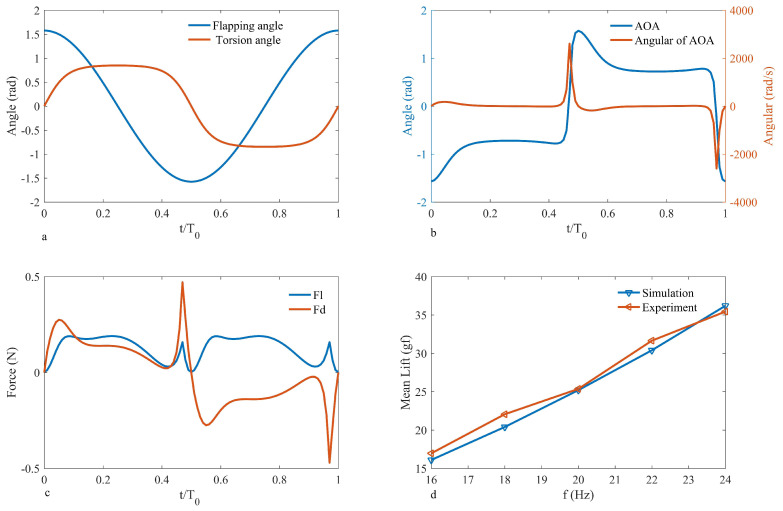
Lift simulation and test curves: (**a**) single-cycle curves of flapping angle and torsion angle; (**b**) single-cycle curves of angle of attack (AOA) and angular of AOA; (**c**) single-cycle lift and drag curves; (**d**) relationship between average lift and frequency. Comparison of simulation results with experimental results.

**Figure 4 biomimetics-09-00449-f004:**
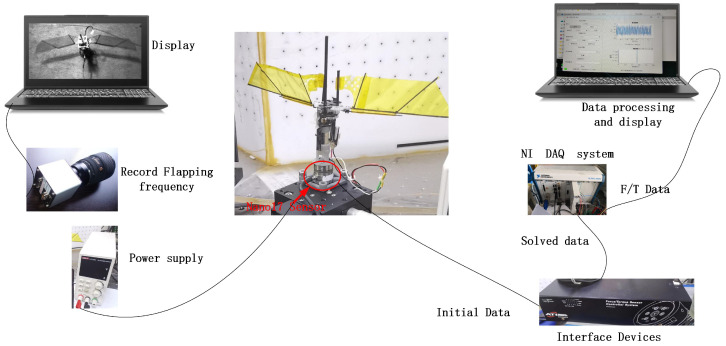
Lift testing platform diagram.

**Figure 5 biomimetics-09-00449-f005:**
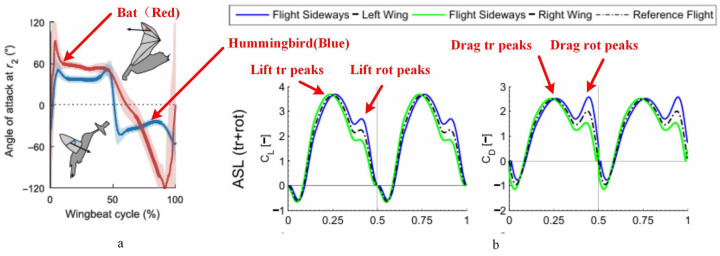
References to other works in the simulation model: (**a**) angle of attack patterns observed by Rivers in hummingbird motion (blue) [34]; (**b**) Matej’s lift and drag coefficient simulation results [29].

**Figure 6 biomimetics-09-00449-f006:**
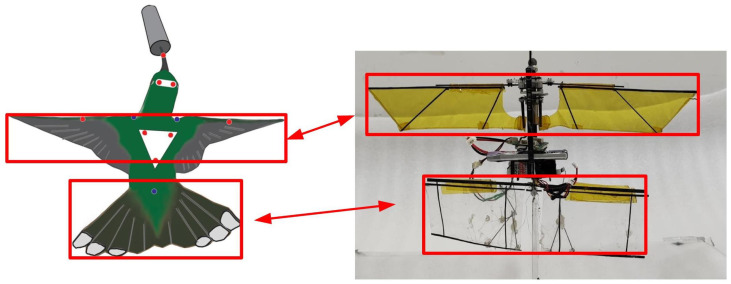
Comparison between the wing and tail area ratios of Archilochus colubris [38] and the FMAV in this paper.

**Figure 7 biomimetics-09-00449-f007:**
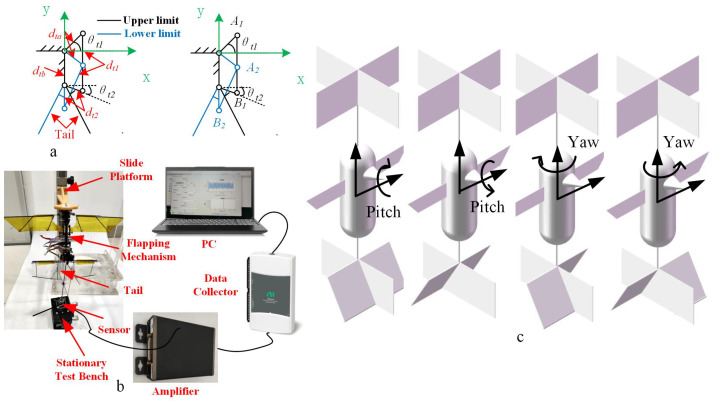
Tail experiment. (**a**) Schematic diagram of tail parameter design. (**b**) Tail moment test bench. (**c**) Direction of the torque generated by tail rotation.

**Figure 8 biomimetics-09-00449-f008:**
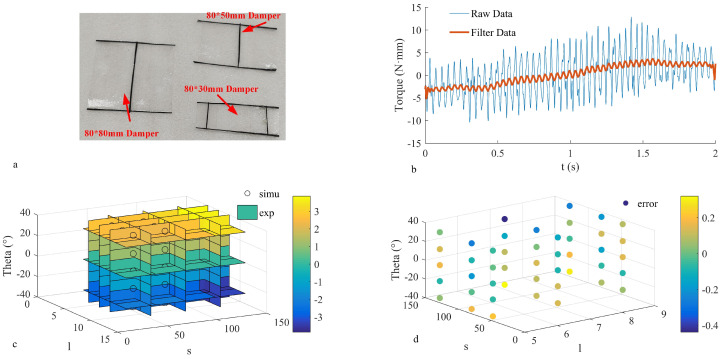
Tail torque measurement results. (**a**) Actual tail diagrams of different sizes. (**b**) S = 8 cm × 5 cm tail, l = 5 cm torque test chart. (**c**) Comparison of tail torque experimental results and simulation results. (**d**) Error value between tail test points and simulation results.

**Figure 9 biomimetics-09-00449-f009:**
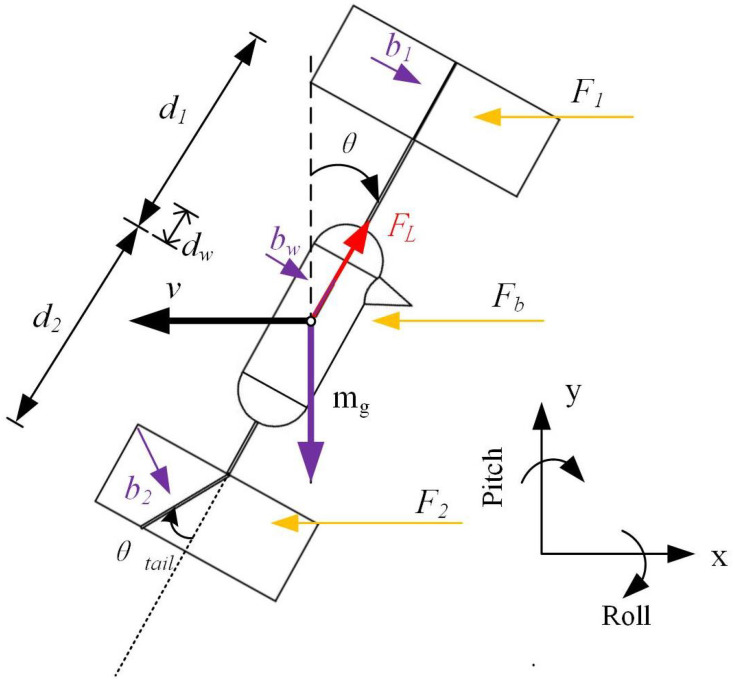
Resistance model of variable-tail FMAV system.

**Figure 10 biomimetics-09-00449-f010:**
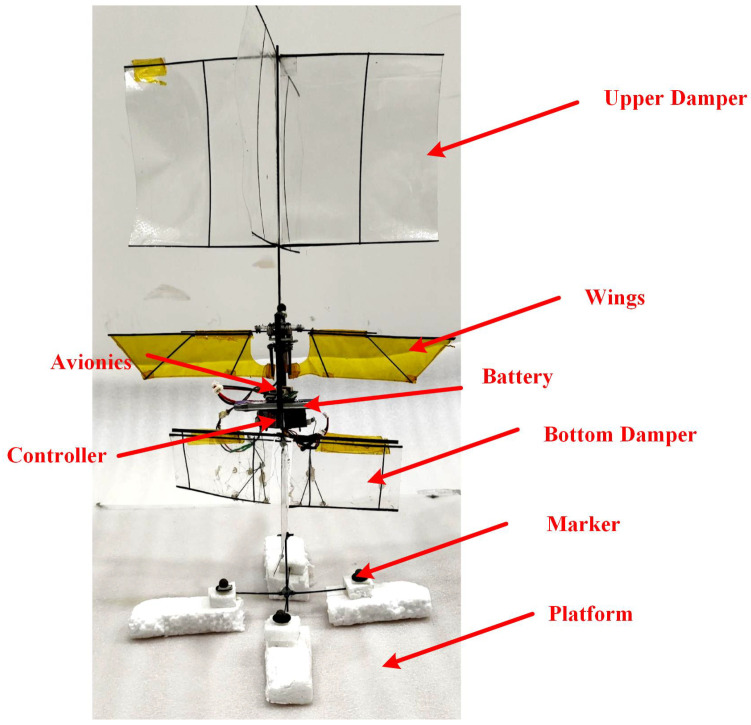
Physical prototype.

**Figure 11 biomimetics-09-00449-f011:**
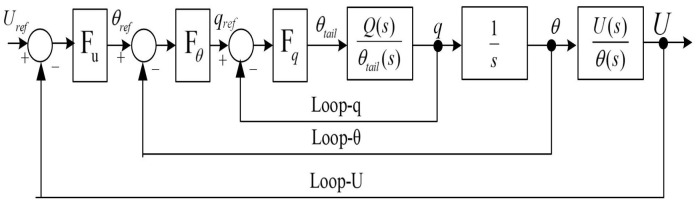
Emulating the controller framework.

**Figure 12 biomimetics-09-00449-f012:**
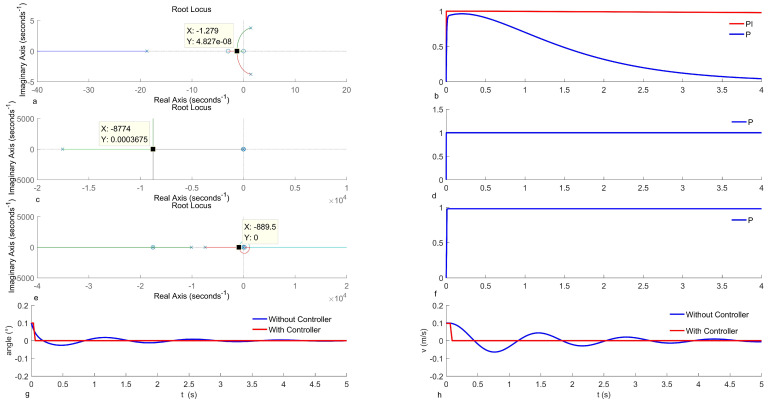
Controller simulation experiments. (**a**) Angular velocity adjustment root trajectory curve; (**b**) angular velocity proportional–integral controller adjustment results; (**c**) angle adjustment root trajectory curve; (**d**) angle proportional controller adjustment results; (**e**) velocity proportional controller root trajectory curve; (**f**) velocity proportional controller adjustment results; (**g**) comparison of the original model and the applied controller angular response; (**h**) comparison of the original model and the applied controller velocity response.

**Figure 13 biomimetics-09-00449-f013:**
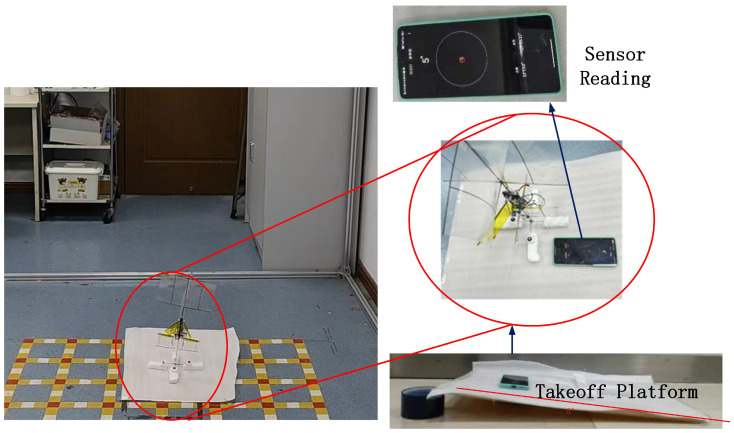
The take-off test bench of the physical prototype.

**Figure 14 biomimetics-09-00449-f014:**
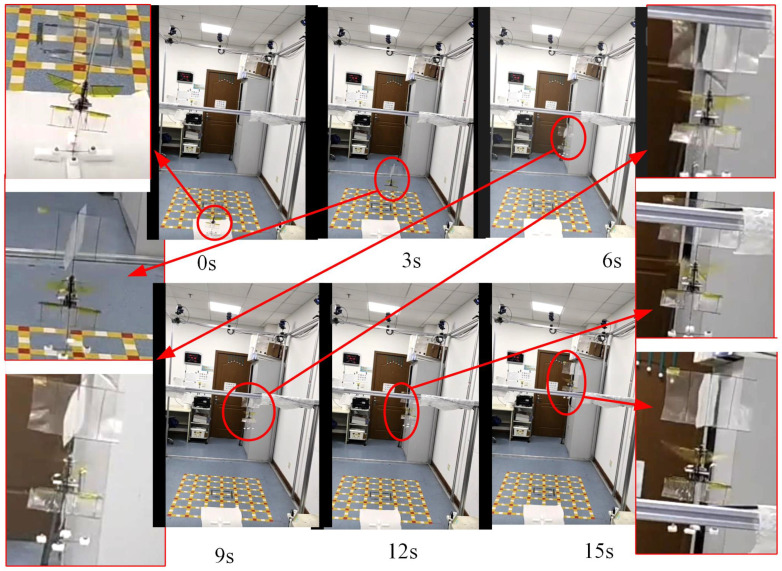
Images of the 0° angle of attack flight experiment.

**Figure 15 biomimetics-09-00449-f015:**
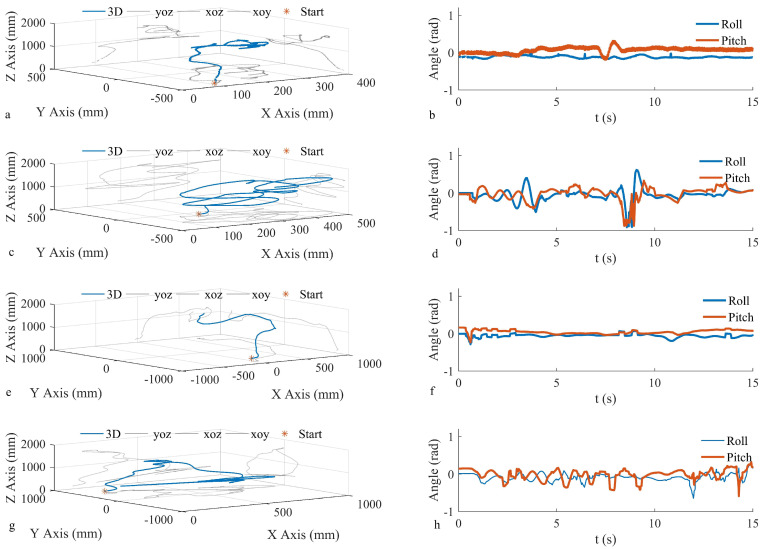
Prototype take-off test. (**a**) The initial take-off 3D trajectory of the prototype with tilt and feedback controller. (**b**) The initial non-tilt of the prototype with feedback controller during take-off; pitch axis and roll axis recording. (**c**) The initial take-off 3D trajectory of the prototype without tilt and feedback controller. (**d**) The initial non-tilt and non-feedback controller of the prototype is recorded on the pitch axis and roll axis during take-off. (**e**) The 3D trajectory of the take-off of the prototype with a feedback controller at the initial 5° tilt. (**f**) The initial 5° tilt of the prototype has a feedback controller recording the pitch axis and roll axis during flight. (**g**) The 3D trajectory of the take-off of the prototype without a feedback controller at the initial 5° tilt. (**h**) At the initial 5° tilt, the pitch axis and roll axis are recorded during the take-off of the prototype without feedback controller.

**Table 1 biomimetics-09-00449-t001:** Comparison of lift with parameters from natural organisms.

Parameter	Wing Length, *R* (mm)	Mean Chord, $\bar{c}$ (mm)	Torsion Angle, $\alpha_{0}$ (°)	Frequency, *f* (Hz)	Sweep Amplitude, $A_{m}$ (°)	Lift (This Paper) (gf)	Lift (CFD) (gf)
Fruit flies [30]	2.39	0.9	0	218	140	$8.8\times 10^{-3}$	$9.6\times 10^{-3}$
*Episyrphus* *baltealus* [31]	9.7	2.26	−34.4	162	65.6	0.225	0.186
*Agrius* *convolvuli* [32]	48.3	18.3	−15	26.1	115	17.01	17.0

**Table 2 biomimetics-09-00449-t002:** Calibration of measurement sensor.

Weight (N)	1	2	5
Value (N)	1.0015	1.9979	5.0005
Error (%)	0.15	−0.15	0.01

**Table 3 biomimetics-09-00449-t003:** Comparison data of tail and wing dimensions.

Parameter	Tail Width (cm)	Wingspan (cm)	Width-Span Ratio	Tail Length (cm)	Wing Chord (cm)	Length–Chord Ratio
Teoh [9]	1.5	3	0.5	1.5	1.3	1.154
Teoh [9]	2	3	0.667	2	1.3	1.538
Breugel [8]	15	21	0.714	15	4	3.75
Altartouri [10]	8	21	0.31	6	2.5	2.4

**Table 4 biomimetics-09-00449-t004:** Damper experimental data.

Parameter	$\theta_{t1}$	$\theta_{t2}$	$d_{\mathrm{ta}}$	$d_{\mathrm{tb}}$
Value	(−45°, 45°)	(20°, 70°)	0.5 cm	2 cm

**Table 5 biomimetics-09-00449-t005:** Table of prototype parameters.

Name	Parameter	Value
Body mass (g)	m	25
Distance from the center		
of mass to the lift point (mm)	$d_{w}$	10
Moment of inertia of the body (g·mm^2^)	J	136
Flapping damping (Ns·m^−1^)	$b_{w}$	$3.28\times 10^{-2}$
Area of the top damper (cm^2^)	$S_{1}$	250
Distance from the top damper to		
the center of gravity (mm)	$d_{1}$	140
Moment of inertia of the top damper (g·mm^2^)	J	$3.89\times 10^{4}$
Damping of the top damper (N·s·m^−1^)	$b_{1}$	$1.37\times 10^{-2}$
Area of the bottom damper (cm^2^)	$S_{2}$	64
Distance from the bottom damper		
to the center of gravity (mm)	$d_{2}$	80
Damping of the bottom damper (N·s·m^−1^)	$b_{2}$	$2.74\times 10^{-2}$
Wing–tail separation (mm)	l	76

**Table 6 biomimetics-09-00449-t006:** Model’s parameter values.

Parameter	$X_{u}^{+}$	$X_{q}^{+}$	$M_{u}^{+}$	$M_{q}^{+}$	$N_{\alpha }^{+}$
Value	−2.98	−0.044	−31.11	−12.9	25.05

**Table 7 biomimetics-09-00449-t007:** Statistical results of mean and standard deviation for take-off at 0° and 5° in previous work.

Direction		Pitch		Roll
Value	Mean	Standard deviation	Mean	Standard deviation
	(rad)	(rad)	(rad)	(rad)
Take-off at 0 degrees	0.0412	0.0948	0.167	0.187
	−0.0409	0.0992	0.096	0.136
	0.0172	0.1384	0.0751	0.159
Take-off at 5 degrees	0.0054	0.1461	0.163	0.181
	−0.0024	0.1896	0.0913	0.211
	−0.0714	0.1338	0.0243	0.136
	−0.0742	0.0352	0.0907	0.108

## Data Availability

The data are available from the corresponding author upon reasonable request.

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
