# Peer review of "Stability and Controller Research of Double-Wing FMAV System Based on Controllable Tail"

_biomimetics, 2024, doi:10.3390/biomimetics9080449_

Round 1

Reviewer 1 Report

Comments and Suggestions for Authors

The paper explores the design of a double-wing FMAV system with a controllable tail to enhance stability and control. By introducing a controllable tail, the paper aims to improve the aircraft's stability and control capabilities, contributing to advancements in aerial vehicle technologies. However, there are some contents in this manuscript that lack clarity. It is hoped that these can be addressed in the revised manuscript.

1. What is the “biomimetic” point of this design? Although it mentions that birds use their tails for stability and control, the way the model's tail operates seems to differ from that of birds. In fact, the entire article does not use terms such as 'biomimetic' or 'bioinspired' at all.
2. Although the dimensions of the main wing do not be mentioned in the manuscript, the tail wing appears to be disproportionately large in comparison to the main wing based on the pictures shown in the manuscript. What is the reason behind this design choice? This seems somewhat disproportionate compared to the published FMAVs.
3. The caption of Figs. 11(a) and (c) are confusing (possibly typos?). Although some clarifications are provided in the text, the paper derives the equations of motion without explaining the parameters used in the control system itself, thus limiting the information available to the readers.

4. This paper provides extensive descriptions of the control system and model design; however, it relatively lacks presentation of the results. The flight test results in Figure 11 are not easily interpretable from the graph alone, and the trajectory plots appear somewhat confusing. It might be beneficial to include appropriate pictures to better present the flight test results.

Besides, the research may not cover comparisons with other existing FMAVs and the stability enhancement methods or technologies in the field of biomimetics. Possible limitations in the experimental validation of the controllable tail system's effectiveness and applicability should be investigated or, at least, be discussed in this manuscript. 

Comments on the Quality of English Language

The readability of the English content is not an issue

Reviewer 2 Report

Comments and Suggestions for Authors

Animal flyers like birds produce tail control torques to maintain their attitude. This mechanism is termed as wing-tail interaction. This study aims to apply this mechanism to the control of a flapping wing aircraft.

Major comments:

1. One of major claims regarding the novelty in this study is “the prototypes lack control mechanisms to adjust the oscillations and disturbances generated during flight”. However, there exit many studies that introduce tail control to use wing-tail interaction. Hence, the reviewer does not understand the scientific contribution of this study. In this manuscript, the authors just implemented the control of the tail in their own prototype. This understanding comes from not only poor presentation of the novelty, but also poor presentation of the results and discussion. For examples, the authors did not compare their numerical and experimental results with the published data at all. As one important example, the effect of the proposed tail control to the flight performance improvement is compared with only their own previous result (without tail control). The flight performance improvement must be compared with other published data.

2. Figure 3a: Lift and drag force histories are quite different from typical time histories given by flapping wings. The authors must discuss the validity of these histories and they must compare these histories with those in the previous studies.

3. Figure 3b: The simulation results must be compared with any published data. If the authors compare them with their own experimental results, they must describe the detail of the experiment such that the readers can replicate the experimental results.

4. Figure 5b: Why the unit of torque is degree? In addition, there is no sensor information such as the measurement principle, the instrument’s name, the measurement accuracy, etc. What filter is used? Why the raw data is not used for discussion? These questions are come from because the review does not observe any noise in the raw data, which should be removed.

5. In all experimental results, the measurement accuracy must be presented. Of course, the details of the experimental methods must be adequately presented such that the reader can replicate the experiment.

6. There exist too many typos, mistakes, and unusual expressions regarding scientific writing. The scientific writing quality is too poor to consider the publication.

Minor comments:

1. The keywords are not used or not defined in the main body of the manuscript. Please use or define them in the main body.

2. In Figure 5c, the reviewer cannot understand why the experimental data are continuous in each plane.

Comments on the Quality of English Language

There are too many typos.

Author Response

Please see the attachment."

Reviewer 3 Report

Comments and Suggestions for Authors

The topic is interesting. However, some comments should be revised. The detailed comments are as follows:

1.      What specific factors and parameters were considered in developing the stability model for the double-wing FMAV? How were these parameters quantified and validated?

2.       Is the model for calculating the lift force of the flapping wings a novel development, or is it based on existing models? If based on existing models, what modifications or improvements were made?

Discussion on the proposed law could benefit from a recently reported reference e.g. https://doi.org/10.1016/j.ins.2023.120087

How were the response times for angle and velocity measured? What were the conditions under which the reduction from 4s to 0.1s and 5.64s to 0.1s were achieved?

How many take-off experiments were conducted to conclude the 72.96% and 56.85% reductions in standard deviation? Were these experiments performed under controlled conditions?

What metrics were used to compare flight stability between the passive stabilized prototype with an adjustable tail angle and the fixed tail designs? How consistent were these metrics across multiple trials?

 What are the potential reasons behind the significant reduction in angular deflection and improvement in flight stability due to the tail angle controller? Are there any theoretical underpinnings that support these results?

 How generalizable are the findings to other types of flapping micro air vehicles? Can the feedback-controlled tail design be adapted for different FMAV configurations?

 What further tests or validations do you plan to conduct to reinforce the findings? Are there plans to test the FMAV in more varied environmental conditions?

How might the control system be further optimized to enhance the performance of the FMAV? Are there other parameters that could be fine-tuned for better results?

How was the 2.4% error between the lift force model and experimental data determined? What was the sample size and conditions of the experiments used for validation?

Include limitations of the control law in conclusions.

Some figures, especially the results/graphs, are of low quality. This could have happened due to conversion in PDF by the submission system. Please check it.

Thoroughly proofread the paper for typos and linguistic improvements.

Make sure that all abbreviations are elaborated/defined.

Comments on the Quality of English Language

The topic is interesting. However, some minor comments should be revised. The detailed comments are as follows:

·         What specific factors and parameters were considered in developing the stability model for the double-wing FMAV? How were these parameters quantified and validated?

·         Is the model for calculating the lift force of the flapping wings a novel development, or is it based on existing models? If based on existing models, what modifications or improvements were made?

·         Discussion on the proposed law could benefit from a recently reported reference e.g. https://doi.org/10.1016/j.ins.2023.120087

·         How were the response times for angle and velocity measured? What were the conditions under which the reduction from 4s to 0.1s and 5.64s to 0.1s were achieved?

·         How many take-off experiments were conducted to conclude the 72.96% and 56.85% reductions in standard deviation? Were these experiments performed under controlled conditions?

·         What metrics were used to compare flight stability between the passive stabilized prototype with an adjustable tail angle and the fixed tail designs? How consistent were these metrics across multiple trials?

·         What are the potential reasons behind the significant reduction in angular deflection and improvement in flight stability due to the tail angle controller? Are there any theoretical underpinnings that support these results?

·         How generalizable are the findings to other types of flapping micro air vehicles? Can the feedback-controlled tail design be adapted for different FMAV configurations?

·         What further tests or validations do you plan to conduct to reinforce the findings? Are there plans to test the FMAV in more varied environmental conditions?

·         How might the control system be further optimized to enhance the performance of the FMAV? Are there other parameters that could be fine-tuned for better results?

·         How was the 2.4% error between the lift force model and experimental data determined? What was the sample size and conditions of the experiments used for validation?

·         Include limitations of the control law in conclusions.

·         Some figures, especially the results/graphs, are of low quality. This could have happened due to conversion in PDF by the submission system. Please check it.

·         Thoroughly proofread the paper for typos and linguistic improvements.

·         Make sure that all abbreviations are elaborated/defined.

Round 2

Reviewer 1 Report

Comments and Suggestions for Authors

The author has made good revisions based on previous review comments

Author Response

Comment 1: [The author has made good revisions based on previous review comments]

Response 1: Thank you!

Reviewer 2 Report

Comments and Suggestions for Authors

The reviewer considered all comments have been addressed by the authors sufficiently except the following point:

The authors have not addressed sufficiently the following comment of the first-round:

----------------

2. Figure 3a: Lift and drag force histories are quite different from typical time histories given by flapping wings. The authors must discuss the validity of these histories and they must compare these histories with those in the previous studies.

----------------

Your force histories have sharp peaks in the stroke reversals (t/To = 0.5, 1.0, …). The reviewer consider these unnatural characteristics have not been presented in the typical force histories in other literatures about small animal flyers. Could you discuss about this point?

Author Response

Dear Reviewer:

Thank you for your comments concerning our manuscript entitled “Stability and Controller Research of double-wing FMAV System Based on Controllable Tail” (biomimetics-3071965). Those comments are all valuable and very helpful for revising and improving our paper, and also have important guiding significance to our research. We have studied comments carefully and made corrections which we hope meet with approval. Revised portions are marked in red in the manuscript and this document. 

Reviewer 3 Report

Comments and Suggestions for Authors

All concerns are solved 

Author Response

Comment 1: [All concerns are solved]

Response 1: Thank you!